# *In situ* characterization of nanoparticle biomolecular interactions in complex biological media by flow cytometry

Maria Cristina Lo Giudice[1], Luciana M. Herda[1], Ester Polo[1] & Kenneth A. Dawson[1]

Nanoparticles interacting with, or derived from, living organisms are almost invariably coated in a variety of biomolecules presented in complex biological milieu, which produce a bio-interface or 'biomolecular corona' conferring a biological identity to the particle. Biomolecules at the surface of the nanoparticle–biomolecule complex present molecular fragments that may be recognized by receptors of cells or biological barriers, potentially engaging with different biological pathways. Here we demonstrate that using intense fluorescent reporter binders, in this case antibodies bound to quantum dots, we can map out the availability of such recognition fragments, allowing for a rapid and meaningful biological characterization. The application in microfluidic flow, in small detection volumes, with appropriate thresholding of the detection allows the study of even complex nanoparticles in realistic biological milieu, with the emerging prospect of making direct connection to conditions of cell level and *in vivo* experiments.

[1] Centre for BioNano Interactions, School of Chemistry and Chemical Biology, University College Dublin, Belfield, Dublin 4, Ireland. Correspondence and requests for materials should be addressed to E.P. (email: ester.polotobajas@cbni.ucd.ie) or to K.A.D. (email: Kenneth.A.Dawson@cbni.ucd.ie).

Interactions between engineered nanoscale constructs and living organisms are mediated by an interface at which biomolecules, both chemically grafted and accreted from the biological environment, substantially modify the bare material interface[1–4]. Recently, evidences in the literature have shown a strong correlation between the nature of this complex multilayer of biomolecules (often called biomolecular corona) and the cellular uptake of nanoparticles *in vitro* and *in vivo*[5–13]. In this respect a key role is played by the organization and mutual orientation of the molecules on the nanoparticle surface. While it can be certainly hypothesized that the exposure of certain protein domains on the nanoparticles corona can trigger specific cellular recognition pathways, the environment in which the biological recognition occurs plays a key role in the recognition event itself and has to be taken into account. Cell ligand interactions typically involve receptor engagement with 'recognition motifs', that is, specific molecular structural elements expressed at the nanoparticle surface[14–17]. However, information on the presentation of such recognition motifs on this complex and highly dynamic multilayer of biomolecules, required to understand and design bionanoscale objects, is not easily or widely accessible, limiting our ability to develop a mechanistic understanding of the field.

Current methodologies require the isolation of the nanoparticle–biomolecular corona complex from the biological environment and the subsequent characterization in different media, typically phosphate saline buffer (PBS) or water, often accomplished through more or less harsh treatments that can further modify the nanoparticle bio-interface[18–21]. Moreover the average compositional information obtained with the current techniques does not fully account for the complexity of the nanoparticle–corona–cellular receptor interactions. Both widely available and increasingly sophisticated methods will be required to characterize the molecular details on the surface of nanoparticles as they are made, and modified *in situ* in the presence of the biomolecules from the environment in which they are exposed. It is therefore imperative to seek for methodologies that enable to acquire molecular information in a realistic biological scenario.

Here we introduce a flow cytometry-based methodology that allows for the detection of molecular motifs presented for biological recognition on the nanoparticle surface, in simple and highly complex dispersions and biological milieu. Thereby, by gaining structural characterization of the composition and organization of biomolecules on the nanoparticle surface, we clarify the nanoparticle biological identity, and may hypothesize receptor engagements, and therefore the nanoparticle biological impact.

Our approach is based on the use of highly specific reporter binders, in the present case antibodies (Ab) conjugated to quantum dots (QDAb), that target recognition sites proximate to receptor binding sites. QDs possess high absorption cross-sections across all wavelength ranges, high levels of brightness and photo-stability, and narrow emission bandwidths, allowing for multiple simultaneous labelling and detection of different colours associated with different recognition centres[22–26]. After the nanoparticles have been titrated with these QDAb reporters, their detection in microfluidic flow in principle allows for multiple and simultaneous detection of small groups or even individual particles allowing for analysis of nanoparticle bio-interfaces[27].

Here we show that routine flow cytometers intended for cell analysis, available in most biology laboratories, enable a qualitative and some semi-quantitative understanding of the nanoparticle bio-interface[28]. For bio-interface mapping, QDAb are titrated against dispersions of nanoparticles presenting a biomolecular corona until all accessible target sites are exhausted and measurements of scattering and fluorescence take place in a very small detection volume. The detector threshold may be arranged to eliminate background from unbound labels, and scattering from the complex dispersion medium is significantly reduced. This methodology enables the characterization of the specific motifs of biomolecular corona, allowing to elucidate and ultimately predict nanoparticle biomolecular interactions with cells.

## Results

**Flow cytometry analysis of single protein–nanoparticle model.**
For validation we use a single protein–nanoparticle model, first removing the excess of unbound QDAb and comparing the results from flow cytometry and steady state fluorescence spectroscopy. Dispersions of 200 nm non-fluorescent polystyrene (PS) nanoparticles with a single adsorbed protein layer of human Transferrin (Tf) forming complexes (PS@Tf nanoparticles) were characterized using differential centrifugal sedimentation (DCS), dynamic light scattering (DLS) and nanotracking (NTA) analysis (Supplementary Fig. 1; Supplementary Tables 1 and 2). Highly luminescent water soluble CdTe QDs with tunable core sizes modulating fluorescence emission band (Supplementary Fig. 2) are used. In the current example 4 nm QDs conjugated to a monoclonal (m) antibody that recognizes Tf epitope AA142-144 ($mTfQD_{630}$) allows us to recognize sites close to the Tf receptor binding site (Fig. 1b; Supplementary Fig. 3).

The lower size detection limit for light scattering of conventional widely available flow cytometers is typically of order 200–500 nm, though fluorescence measurements are more sensitive. A suitable compromise involves higher concentrations (termed the 'swarm regime')[29,30] in which multiple nanoparticles, captured within the detection volume, are simultaneously illuminated by the laser and counted as a single event (Fig. 1). To establish the experimental set up, a systematic variation of the nanoparticle concentration (of size ranging from 200 to 50 nm) was studied by flow cytometry, to determine a suitable nanoparticle concentration that overcomes the background noise (Supplementary Figs 4–7). Here a concentration of 0.5 mg ml$^{-1}$ for 200 nm polystyrene nanoparticles ($8.5 \times 10^{10}$ number of nanoparticles $\cdot$ ml$^{-1}$, determined by NTA (Supplementary Table 1), generates a distinctive side scattering signal from the background (Supplementary Fig. 4b) and for the particular configuration used here (see specifications in Fig. 1a) this concentration corresponds to 3,000 nanoparticles present in the quartz capillary of the instrument under laminar flow, of which 150 nanoparticles (on average, in a range of 100–225 nanoparticles) are illuminated simultaneously by the laser at each event generation (see sketch inset in Fig. 1a).

Fluorescence (vertical axis) and high angle (90°, SSC-A) scattering area event distributions were recorded (Fig. 2a) along the titration curve of added QDAb (see controls in Supplementary Figs 8 and 9). The population-mean fluorescence signal (Fig. 2b) is correlated to the amount of immuno-QDs present on the nanoparticle surface, and thereby to specific Tf epitopes on the nanoparticle surface (Fig. 2c). The increase in the side scattering signal (Fig. 2a) is also, as expected, correlated with the ligation of the QDAb on the particle surface, and heterogeneity in the quality of dispersion is also readily detected.

For both fluorescence and scattering the titration is carried to saturation at which point no further target sites are accessible to QDAb (Fig. 2c) for 200 nm (triangles) and 100 nm (open circles) polystyrene nanoparticles. This approach gives a rapid and practical assessment of the quality of both dispersion and available epitopes in a more complete biological context. Semi-quantitative absolute number estimates of epitopes may be derived from a calibration curve linking the mean fluorescence in the detection volume in the flow cytometer to fluorimeter

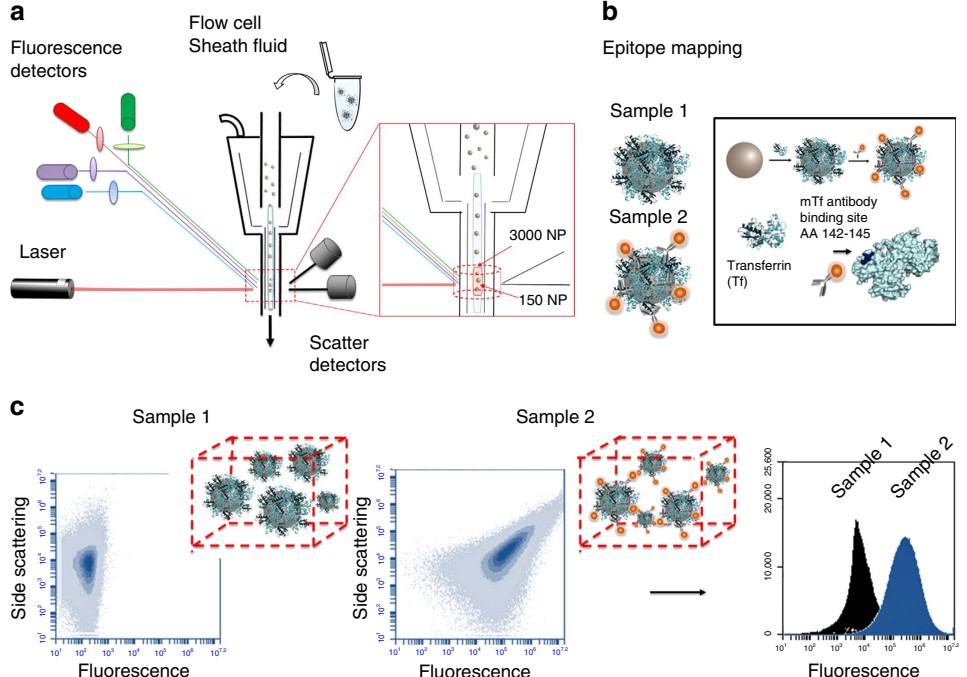

**Figure 1 | Schematic representation of epitope mapping by flow cytometry.** (**a**) Graphical representation of the flow cytometry analysis of nanoparticle samples, illustrating the 'swarm effect'. Flow cytometry allows to analyse nanoparticle dispersion under microfluidic laminar flow conditions. By increasing the number of nanoparticles simultaneously illuminated by the laser in the detection volume, the average scattering properties of the media change. As a consequence the signal-to-noise ratio in the side scattering channel increases enabling the distinction of the signal due to multiple nanoparticles ('swarm') from the instrumental background. (**b**) QDs functionalized with specific monoclonal antibodies used for fluorescent labelling to map out target relevant epitopes of the protein corona, by titration of the label against the nanoparticles. (**c**) Flow cytometry allows fluorescence measurements of very small groups of particles (swarm regime) coupled to simultaneous measurements of low angle (forward) and high angle (side) light scattering. In principle this provides simultaneous information on the local state of nanoparticle dispersion, and the fluorescence per particle labelled with QDAb.

intensity (Fig. 2d; Supplementary Fig. 10). Here we find the total number of Tf labelled by $mTfQD_{630}$ is $364 \pm 27$ epitopes for 200 nm polystyrene nanoparticles (and $558 \pm 57$ epitopes for polyclonal antibody $pTfQD_{630}$, see Supplementary Fig. 11), and $145 \pm 10$ epitopes for 100 nm polystyrene nanoparticles, in agreement with previous estimates obtained from other methods[31]. These numbers of epitopes available correspond to 24 and 30% of the total amount of protein determined by Bicinchoninic acid assay (BCA) (Supplementary Fig. 12) for 200 and 100 nm polystyrene nanoparticles, respectively. Variations of signals from bound QDAb derived from the sample preparation (nanoparticle, nanoparticle-protein complexes, mixing, and so on) are typically small, and the accuracy of the estimated numbers depends on errors accumulated in the calibration process.

The measurement of both fluorescence and side scattering provides simultaneous information on the local state of nanoparticle dispersion, and the fluorescence per particle (Supplementary Fig. 13). This is quite different from conventional macroscopic spectroscopy where the average fluorescence could still reflect the number of bound QDs (and therefore target epitopes) but which gives little information on the connection between the averaged signal and the microstructure of small groups of labelled nanoparticles in more complex samples.

**Multiplexing epitope mapping.** Different QDs can be excited by the same laser source so a single laser on the flow cytometer can be used for multicolour experiments allowing several epitopes from the same or different proteins to be labelled with different QDAb and allowing simultaneous analysis of multi-component

protein corona systems[32]. We illustrate here the multiplexed quantification of a two protein model system consisting of Tf and human serum albumin (HSA) adsorbed on 200 nm polystyrene nanoparticles (PS@Tf/HSA), again beginning by removing the excess of immuno-QDs in the sample to validate the experimental system (Fig. 3a). Figure 3b (Supplementary Fig. 14) shows a conventional hard corona analysis in which excess proteins are removed, and those adsorbed to nanoparticles are stripped from the particles and subjected to SDS–polyacrylamide gel electrophoresis (SDS–PAGE). The overall nanoparticle surface contains a mixture of both proteins. In Fig. 3c,d we illustrate the flow cytometry analysis of the PS@Tf/HSA nanoparticles titrated with a 1:1 mixture of both orange anti-Tf QDs ($mTfQD_{630}$) and green anti-HSA QDs ($mHSAQD_{530}$) (see controls in Supplementary Fig. 9 and Supplementary Figs 15 and 16). In Fig. 3c, we present typical scatter density plots for fluorescence in green channel (3c, i–ii) or red channel (3c, iii–iv) versus side scattering, and increase in fluorescence signal is observed for both epitopes. In Fig. 3c(vi) we show an example of the simultaneous detection of the two QDAb binding, in the two fluorescence channels. Using the calibration curve linking the mean fluorescence in the detection volume in the flow cytometer with the number of immuno-QDs for each channel (Supplementary Fig. 10), semi-quantitative analysis of the different epitopes labelled in the mix sample PS@Tf/HSA was performed (Fig. 3d). For the two protein model system (PS@Tf/HSA) a number of $259 \pm 50$ of Tf sequences per nanoparticle and $104 \pm 5$ of HSA were determined. As a control the one protein model for Tf is studied with either green mTfQD or orange mTfQD, or a mixture of the two, all leading to the same determination of epitope numbers ($370 \pm 10$ Tf epitopes with a 1:1 mixture of $mTfQD_{630}$

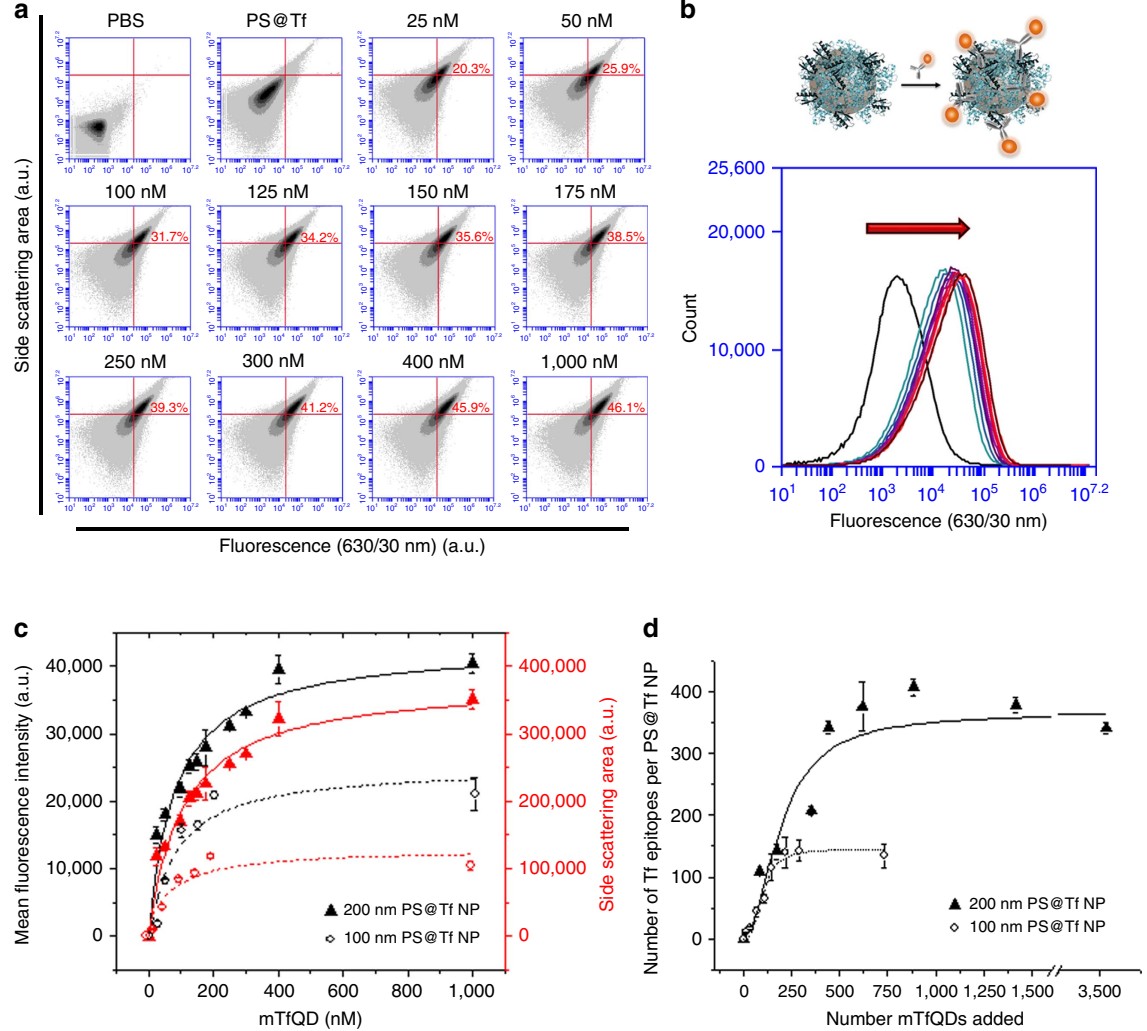

**Figure 2 | Flow cytometry data analysis of PS@Tf nanoparticles labelled with immunoprobes (mTfQD$_{630}$).** (**a**) Scatter density plots of fluorescence signal versus side scattering area of a 200 nm PS@Tf nanoparticle solution (8.5 × 10$^{10}$ PS@Tf nanoparticles · per ml) with increasing mTfQD$_{630}$ concentration. (**b**) Histograms represent the mean fluorescence of each sample of PS@Tf nanoparticle labelled with increasing concentration of immunoprobe mTfQD$_{630}$. (**c**) Fitting analysis of the mean fluorescence signal (in black) and side scattering area signal (in red) of 200 nm PS@Tf nanoparticles (triangles), 8.5 × 10$^{10}$ nanoparticles · per ml, and 100 nm PS@Tf nanoparticles (circles), 8.2 × 10$^{11}$ nanoparticles · per ml, versus the concentration of mTfQD$_{630}$ added. Data represent the mean fluorescence intensity of PS@Tf nanoparticles labelled with QDs ± s.d. of three independent replicates. (**d**) Fitting analysis of the mTfQD$_{630}$ attached to the PS@Tf nanoparticles versus the total number of mTfQD$_{630}$ added. The total number of Tf epitopes labelled by mTfQD in the 200 nm polystyrene nanoparticle is 364 ± 27 and 144.9 ± 10 for the 100 nm polystyrene nanoparticle. Data represent the number of Tf epitopes labelled with mTfQDs ± s.d. of three independent replicates.

and mTfQD$_{530}$, and 364 ± 27, using only mTfQD$_{630}$, see Fig. 3d and Supplementary Fig. 15).

***In situ* epitope mapping of protein–nanoparticle complex.** In reality, bionanoscience, nanosafety and nanomedicine issues all involve nanoconstructs (possibly with pre-attached targeting moieties, grafted or adsorbed as above) further modified by association with various biomolecules adsorbed from the environment in which they are exposed to cells[33–37]. Now we seek structural information about how they are organized and evolve, primarily which recognition fragments are presented for biological interaction, *in milieu* relevant to biology.

First, we repeat the study of Fig. 2 using Tf nanoparticles and mTFQD$_{630}$ but without any washing step (see scheme, Fig. 4a). The mapping analysis with (Fig. 4b, full black curve) and without the free QDAb (Fig. 4b, broken black curve) shows that only QDAb associated to the particles are detectable (Supplementary Figs 9 and 17). When the analysis takes place in a dispersion of

50% human plasma there is a striking reduction of 80% fluorescence from bound QDAb to nanoparticles, and when Tf depleted plasma is used (Supplementary Fig. 18), we find the same result (only about 5% of the total amount of epitopes are detected). No substantial variation in the intensity of the photoluminescence of the QDAb in biological media is observed over the timescale of the experiment (Supplementary Fig. 2).

Furthermore, PS@Tf nanoparticles may be separated after exposure to human plasma (Fig. 4d and Supplementary Fig. 19, one hour and four hours), and new hard corona now isolated on a gel showing that the original Tf layer is modified by secondary adsorption of proteins from the serum.

***In situ* epitope mapping of hard corona nanoparticles.** After exposure to a biological fluid such as serum whether *in vitro* or *in vivo*, individual nanoparticles *in situ* present an inner layer ('hard corona') that is essentially a surface tapestry of different biomolecules and recognition motifs on the same particle[38–41]. In

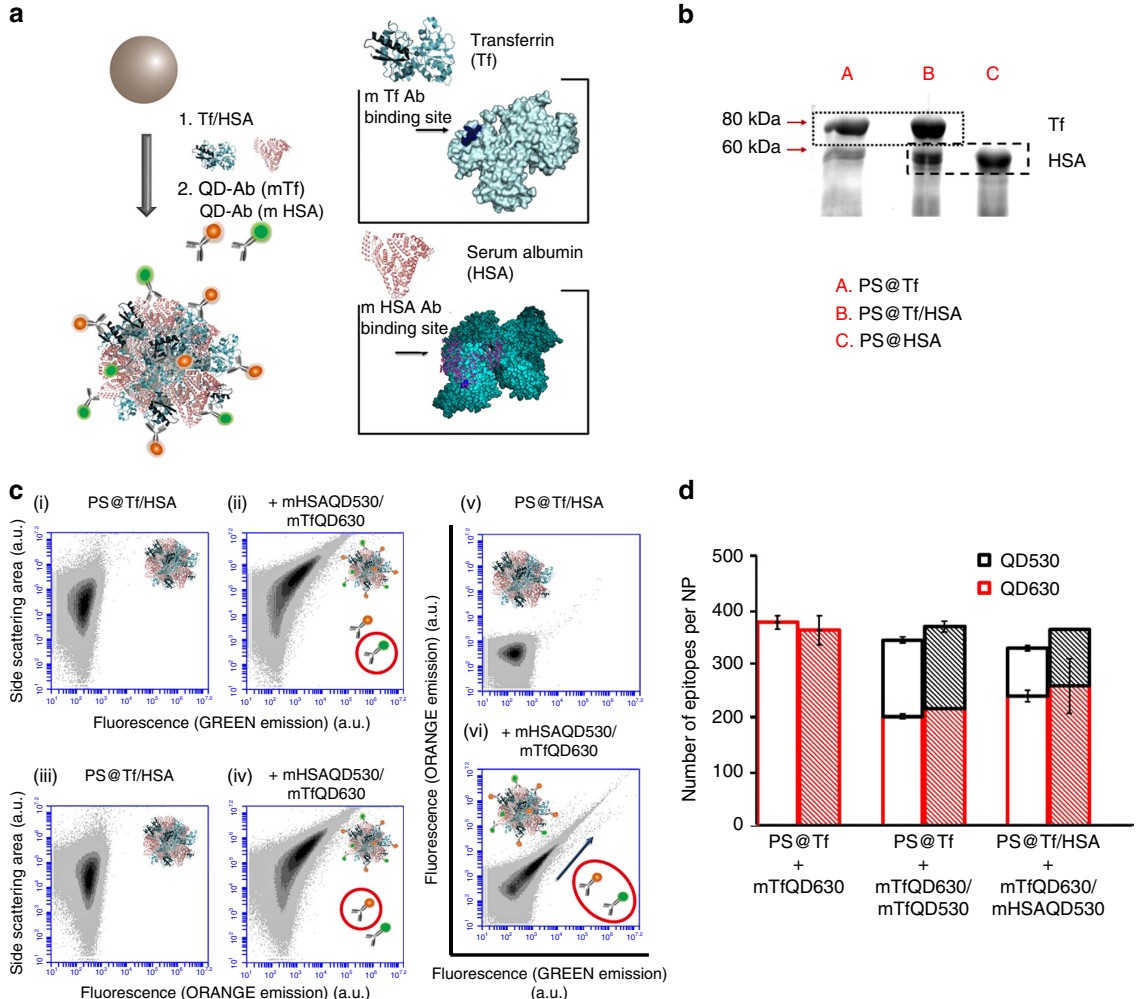

**Figure 3 | Mapping of different epitopes on multi-component protein corona system.** (**a**) Schematic representation of two proteins model system (PS@Tf/HSA nanoparticles) labelled with two different QDs (green and orange QDs, emitting at 530 or 630 nm respectively) functionalized with antibodies anti-HSA and anti-Tf. (**b**) The presence of both proteins adsorbed on the nanoparticle surface was analysed by SDS–PAGE. (**c**) Scatter density plots of fluorescence signal (for green (ii) and orange (iv) emitting QDs) versus side scattering area of a 200 nm PS@Tf/HSA nanoparticles solution ($8.5 \times 10^{10}$ PS@Tf nanoparticles · per ml). Scatter density plots for particles alone (controls) are represented respectively in (i) and (iii). Scatter density plots of fluorescence signal (vi) of a solution of 200 nm PS@Tf/HSA nanoparticles labelled with different QDs (green and orange emitting QDs functionalized with monoclonal antibodies anti-Tf and anti-HSA). Scatter density plots for particles alone (controls) are represented (v). An increase in fluorescence is observed in the sample labelled with mTfQD and mHSAQD. (**d**) Quantitative multiplexing analysis of the number of sequences for a specific protein at the saturation point of each epitope (HSA or Tf immobilized onto polystyrene nanoparticle) by flow cytometry (coloured bars) and fluorescence spectroscopy (empty bars). A sample of PS@Tf nanoparticles where the Tf epitopes are labelled using same mTf antibody conjugated with orange $QD_{630}$ and green $QD_{530}$ has been used as a control. The number of Tf labelled sequences obtained ($370 \pm 10$ Tf per nanoparticle) for the saturation point in the control sample PS@Tf (using $mTfQD_{630}$ and $mTfQD_{530}$), corresponds to the total number of sequences labelled ($364 \pm 27$) by the one single binding model PS@Tf (labelled with $mTfQD_{630}$). A number of $259 \pm 50$ of Tf epitopes per nanoparticle and $104 \pm 5$ of HSA epitopes per nanoparticle were determined for the two protein model (PS@Tf/HSA). Data represent the total number of epitopes labelled with QDs ± s.d. of three independent replicates.

microfluidic flow (for example flow cytometer) the detection volume and detector are such that we can study such systems *in situ* in serum potentially, allowing us to characterize the biological identity, and thereby likely receptor interaction, throughout the duration of an *in vitro* or *in vivo* experiment.

We illustrate these approaches with two types of particles (200 nm polystyrene and 100 nm silica) exposed to 80% human serum (HS) for which hard corona analyses (see Supplementary Table 3) identify abundant Immunoglobulins (IgG) and Apolipoproteins[42,43]. Studying the availability of exposed epitopes of those proteins could potentially predict their interactions with cells respectively via the Fc receptor superfamily or lipoprotein-related receptors (for example, low density lipoprotein receptor recognizes ApoB-100). For the IgG

mapping we use an epitope on the relevant part of the Fc region in the heavy chain (see Supplementary Fig. 3) and results are shown in Fig. 5a. Samples unwashed from QDAb in PBS (black curve in Fig. 5a and black column in Fig. 5b) and samples exposed to IgG depleted human serum (IgGdHS) for one hour (blue curve in Fig. 5a and blue column in Fig. 5b) show epitopes that could interact with Fc receptor are presented throughout the period and conditions of typical cell exposures. Similar conclusions (Fig. 5c) can be drawn for Fc on silica and indeed, for ApoB-100 (Supplementary Fig. 20), suggesting also that those particles could interact with the low-density lipoprotein (LDL) receptor on exposure to cells.

For all of these examples (see for example Fig. 5a,c, red curves) there is a significant reduction of fluorescence (between 85 and

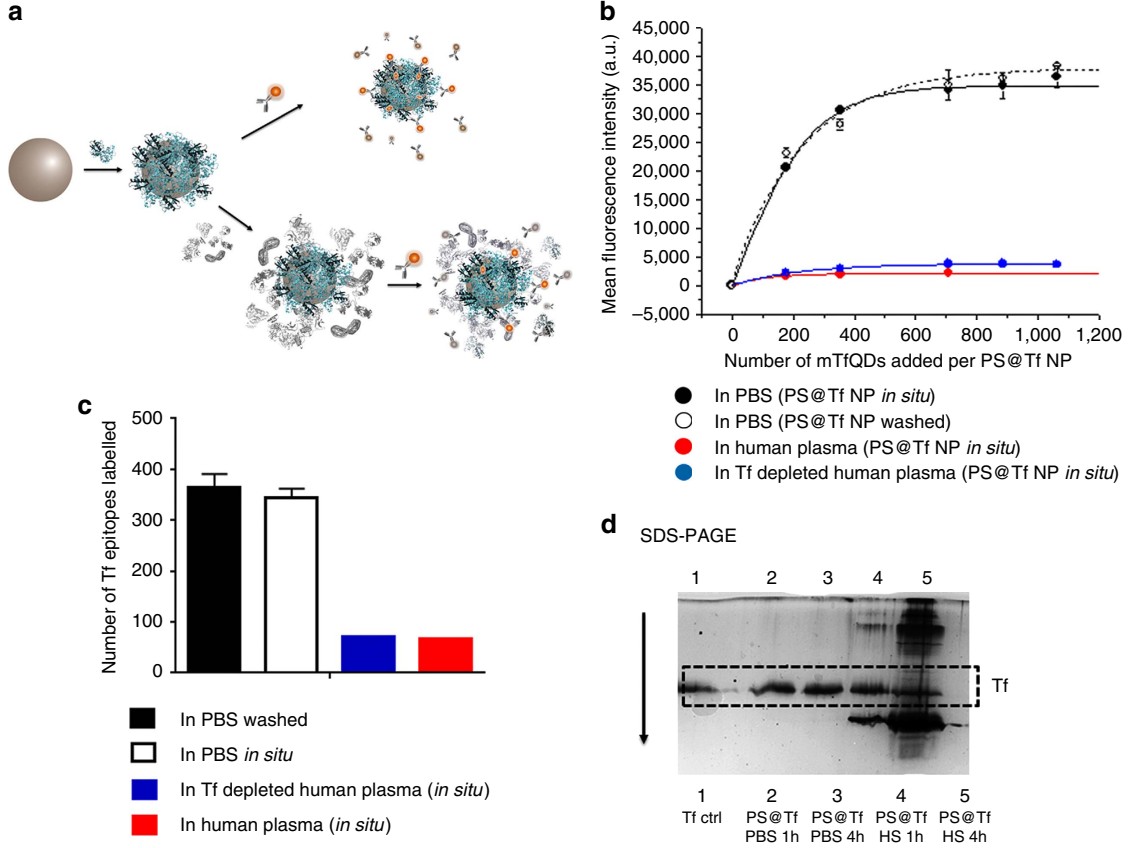

**Figure 4 | *In situ* mapping of Tf epitopes on protein corona of PS@Tf nanoparticle system.** (**a**) Schematic representation of PS@Tf nanoparticles *in situ* labelled with mTfQD$_{630}$ in presence of 50% human plasma and in PBS buffer as a control. (**b**) Flow cytometry data analysis of PS@Tf ($8.5 \times 10^{10}$ PS@Tf nanoparticles · per ml) *in situ* labelled with mTfQD$_{630}$. The samples were analysed without any purification steps in PBS buffer (black line, a control sample after washing the excess of QDs is shown in broken line), human plasma (red line) and Tf depleted human plasma (blue line) after incubation with mTfQD for 1 h. Data represent the mean fluorescence intensity of nanoparticles labelled with QDs ± s.d. of two independent replicates. (**c**) Number of Tf epitopes labelled with mTfQD at the saturation point for the recognition under different media (PBS buffer, human plasma and Tf depleted human plasma). About 5% of the total amount of epitopes are detected after the particles are dispersed in complex media. Data represent the total number of epitopes labelled with QDs ± s.d. of two independent replicates. (**d**) SDS-PAGE analysis of the Tf immobilized on the polystyrene nanoparticle surface when the particles are incubated in PBS buffer or human plasma for 1 and 4 h.

95%) when such samples (whether depleted of the free ligand or not, Fig. 5d) are studied in the presence of the surrounding serum, as would be experienced within living organisms, or in cell studies. For many combinations systems we have studied we find a substantial loss of affinity for the epitope *in situ* (only between 5 and 10% of the total amounts of epitopes are detected). If hard corona complexes are first isolated and then re-dispersed, this occurs over several minutes and the recognition is not much further increased over extended periods of time (Supplementary Fig. 21). We stress that this is not a consequence of the hard corona being changed, and we consider it a consequence of the real nature of such systems. Thus, if the hard corona particles are exposed to high serum concentrations for some hours, separated washed and mapped as usual (Fig. 5a, blue curve), or if the nanoparticle-hard corona particles are first exposed to QDAb, and then subsequently re-dispersed in the same serum media (Fig. 5b, column C) there is no substantial loss of fluorescent intensity. Comparable observations, though all to different degrees depending on the system, are made for wide variety of other systems (silica is presented in Fig. 5c,d).

Combined with numerous other controls (see Fig. 5b,c and Supplementary Figs 2, 22 and 23) we conclude that the antibody mapping is also capable of probing the hard corona 'dressed' or partially screened by the remaining protein in the biological

fluid. The consequences of this effect ('soft corona') have been discussed in several contexts before[44,45], and consequences inferred in a number of situations[46]. However, functional receptors can only rarely be isolated and measurements in realistic media have been limited.

We hypothesize as follows. To identify if proteins present in the macroscopic hard corona (for example see Supplementary Table 3) are in reality likely to interact with receptors, the process of identifying if recognition fragments are presented suitably described here is a pre-requisite. However, numbers of epitopes may alone not be sufficient to rank the avidity for uptake by all the relevant receptors, for additional significant modulation of the actual affinity is likely to arise from the immediate vicinity of tightly bound layer that is the soft corona. This is a substantive point, re-emphasizing the need to work in the presence of real sera in *vitro*. It also suggests the need to further develop our understanding of the multiple origins of the nanoparticle receptor affinity and avidity.

## Discussion

Within molecular cell biology, and the biological and medical sciences more generally, the appreciation of receptor–ligand interactions is highly developed to the point where it is taken for

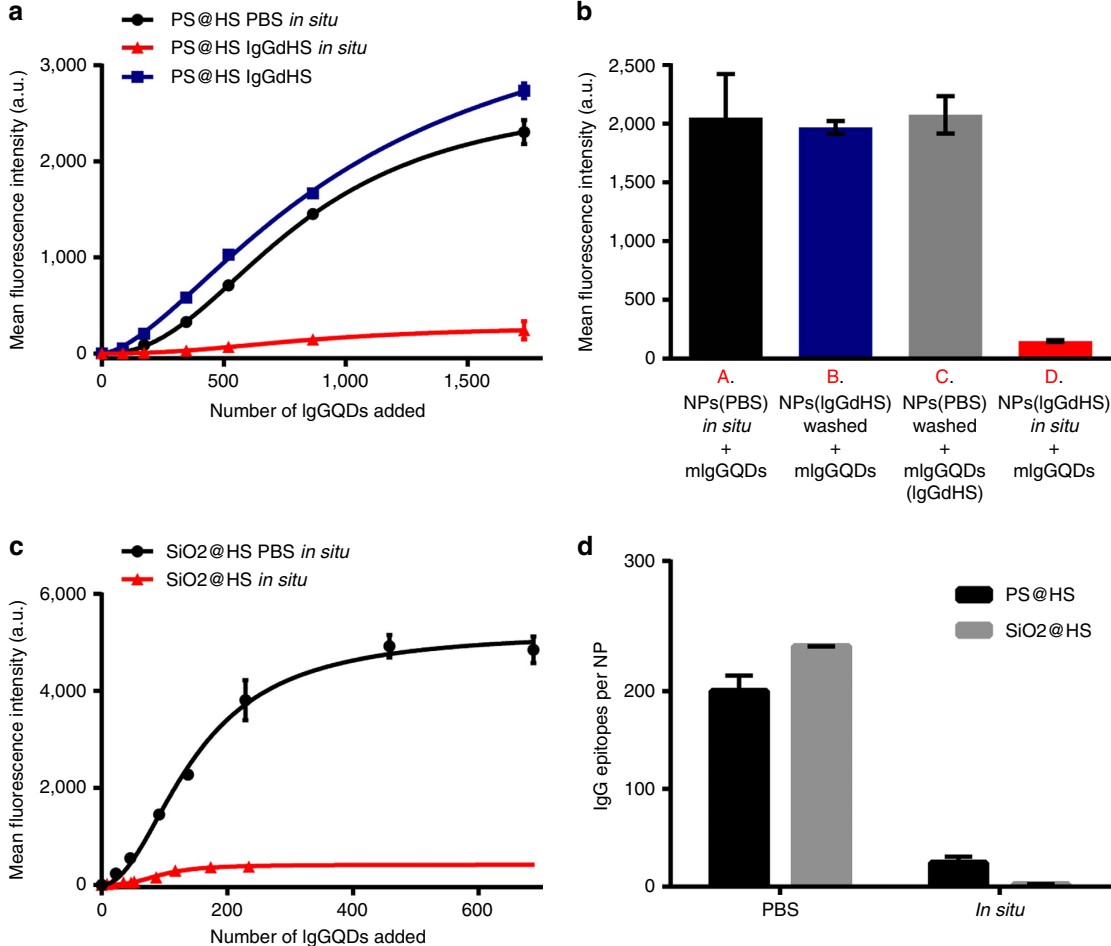

**Figure 5 | *In situ* mapping of IgG (Fc) epitopes on the nanoparticles after formation of a protein corona in 80% human serum (HS).** (**a,b**) Titration curves of polystyrene nanoparticles (PS@HS) with mIgGQD$_{630}$ immunoprobes after the hard-corona nanoparticle complexes were re-dispersed in PBS buffer (black dots and bar) or in a final concentration of 50% IgG depleted human serum (red dots and bar). Data represent the mean fluorescence intensity of nanoparticles labelled with QDs ± s.d. of three independent replicates. The blue dots and blue bar represent PS@HS samples re-dispersed in IgG depleted human serum for 1 h, then spun down and re-suspended in PBS for the *in situ* immunolabelling. These samples showed similar number of labelled epitopes compared with the samples suspended directly in PBS buffer, suggesting that upon 1 h incubation in IgG depleted human serum the IgG epitopes originally present in the corona are not removed by interchange with the proteins of the media. The grey bar represents PS@HS nanoparticles incubated with mIgGQD$_{630}$ in PBS buffer, then washed by centrifugation and re-suspended in 50% IgG depleted human serum before the flow cytometry analysis. The similarity between this result and sample A suggests that the presence of biological media does not influence the output of the flow cytometer. (**c**) Titration curves of silica nanoparticles (SiO$_2$@HS) with mIgGQD$_{630}$ immunoprobes after the hard-corona nanoparticle complexes were re-dispersed in PBS buffer (black) or in a final concentration of 50% IgG depleted human serum (red). Data represent the mean fluorescence intensity of nanoparticles labelled with QDs ± s.d. of two independent replicates. (**d**) Maximum number of IgG epitopes detected in PS@HS and SiO$_2$@HS in PBS or in 50% of IgG depleted human serum. Data represent the total number of epitopes labelled with QDs ± s.d. of two independent replicates.

granted that exquisite levels of molecular details are important in designing and understanding their functional implications in biology. The development of methodologies that can address a molecular level of details on the surface of nanoparticles is now under way.

Here, beginning with propositions on the identity of the proteins at the bio-interface, derived for example from macroscopic proteomics methods, it is possible to determine whether those proteins are actually present, oriented and accessible at the nanoparticle surface. The use of intense fluorescence labels allows even for the identification of relatively rare recognition fragments at the surface of nanoparticles. This type of characterization could rapidly and broadly develop our concept of biologically relevant characterization of nanoparticles.

To deal with all types of nanoparticles (fluorescent and non-fluorescent), multiple nanoparticles are simultaneously illuminated by the laser and captured within the detection volume

(termed the 'swarm regime'), and the outcome counted as one single event. The ability to tune between larger swarms regimes enables the high-throughput profiling of the epitope distribution and the analysis of fluctuations in the nanoparticle population, therefore providing the maximum information on the nature of the ensemble. By using routine instruments as the one employed in this work, which presents limit of detection of 500 nm, we were able to characterize swarms of nanoparticles in the size range of 50 nm. Commercially available instruments equipped with more advanced lasers and robust photodetectors are foreseen to allow the screening of nanoparticles way below this size range. Moreover, the evolution of new microfluidics devices, smaller swarm sizes, and other developments will allow particle by particle evaluation, improving the precision at which absolute numbers of epitopes and their distributions can be determined, and rare epitopes detected, and this will have importance for other complex nanoparticles such as exosomes.

Variations in cellular responses have occasionally been noticed over the years in this field. This observation could potentially be related to different epitope distributions among different batches of nanoparticles due to limited reproducibility of typical preparative procedures for the nanoparticle syntheses, their formulation, or dispersion. The methods outlined here could enable a progressive improvement of the quality of the characterization of the nanoparticle biological functionality and therefore could lead to an increase in the reproducibility in the arena, important for practical medical applications. Much beyond this, though, the use of flow, small detection volumes, and thresholding of detectors combines to allow study of nanoparticles in meaningful scenarios, substantially eliminating background fluorescence from QDAb and scattering from complex milieu, even in realistic concentrations of serum or plasma. This allows us to study the interactions of nanoparticles with cell receptors, *in situ* in the biological media of practical interest. Meaningful characterization of properties *ex vivo* and *in vivo* is now within sight.

These capacities will open the way to the development of the systematic understanding, and consequently rational design of particles and optimization of the early stage interactions with living organisms, with all of the potential consequences for their application to human health.

## Methods

**Chemicals.** All the following chemicals were purchased from Sigma-Aldrich: skimmed milk powder (70,166), 2-(N-Morpholino)ethanesulfonic acid monohydrate MES (69,889), Sodium dodecyl sulfate (L3771), Glycine (G8898), Ammonium persulfate (A3678), Agarose (A9539), Trizma base (T1503), Tween 20 (P1379), Ethylenediaminetetraacetic acid disodium salt dihydrate, EDTA (E4884), N,N,N′, N′-Tetramethylethylenediamine (T9281), Acrylamide/bis-acrylamide,40% solution (A7802), , DL-Dithiothreitol (D5545), Ethanol (32,294–2), Methanol (24,229–2), Phosphate buffered saline PBS tablet (P4417), 3-Mercaptopropionic acid (M5801), Cadmium chloride (202,908), 4-Aminophenyl β-D-galactopyranoside (A9545), Sodium borhydride (S9125), N-(3-Dimethylaminopropyl)-N′-ethylcarbodiimide hydrochloride (E6383), N-Hydroxysulfosuccinimide sodium salt (56,485).Tellurium, 99.8%, powder (315,990,250) was purchased from Acros Organics.

**Nanoparticles.** Plain and carboxylated polystyrene (PS) NPs (20 mg ml$^{-1}$, 100 and 200 nm nominal diameters, Polybead) were purchased by Polysciences Inc. 100 nm nominal diameter fluorescent silica (SiO$_2$) NPs (PSi-G0.1) were obtained from KISKER-BIOTECH. 50 nm fluorescent (Fluorescein isothiocyanate FITC labelled) SiO$_2$ NPs were synthesized following the protocol of Hristov *et al.*[47]

**Proteins.** Holo-Transferrin Human, Tf (T44132), Human Serum Albumin, HSA (A9511) were purchased from Sigma Aldrich. Antibodies: anti-Tf (monoclonal: ab769 and polyclonal: ab9538) monoclonal anti-HSA (ab10241) and monoclonal anti IgG-Fc (ab99770) were purchased from Abcam. Monoclonal antibody anti-Apolipoprotein B-100 (sc-13538) and monoclonal antibody anti-Apolipoprotein E (sc-13521) were purchased from Santa Cruz Biotechnologies.

**Biological fluids.** Human serum (HS) was purchased from BIOCHROM (total protein concentration 77 mg ml$^{-1}$ determined by BCA assay, performed according to manufacturer's instructions). IgG depleted HS (IgGdHS) was purchased from INNOV-RESEARCH (total protein concentration 55 mg ml$^{-1}$ determined by BCA assay, performed according to manufacturer's instructions). Delipidized HS was purchased from SERALAB (total protein content was estimated to be ca. 58 mg ml$^{-1}$ by BCA). Human plasma was obtained by pooling blood samples withdrawn from 10 to 15 different volunteers and prepared following HUPO guidelines[48]. The blood donation procedure was approved by the Human Research Ethics committee at University College Dublin. The final protein concentration was roughly 80 mg ml$^{-1}$, as determined by BCA assay, performed according to manufacturer's instructions.

**Nanoparticle corona complexes.** The nanoparticle corona complexes were prepared fresh before each experiment.

Preparation of polystyrene nanoparticle coated with Tf protein (PS@Tf). 200 or 100 nm PS NPs were incubated with Tf (64 nmol, 5 mg ml$^{-1}$) in MES buffer (pH 6, final NPs concentration 1 mg ml$^{-1}$) at room temperature on a shaker following the procedure described by Kelly *et al.*[31] to saturate the nanoparticle surface. For the preparation of PS@Tf/HSA, 200 nm PS NPs were incubated with

Tf (32 nmol) and of HSA (32 nmol). After 1 h incubation the PS@Tf (PS@Tf/HSA) NPs were centrifuged at 20,000 rcf and re-suspended in fresh buffer to remove the unbound protein. Five washes were performed in total, of which three in MES (pH 6) and two in PBS (pH 7.4). The particles were finally re-suspended in PBS.

Preparation of nanoparticle HS protein corona (PS@HS and SiO$_2$@HS). 200 nm PS NPs and 100 nm SiO$_2$ NPs were incubated with 80% HS in PBS buffer at a final NPs concentration of 1 mg ml$^{-1}$. After 1 h incubation at 37 °C at constant agitation, NPs were washed three times by centrifugation at 20,000 rcf, and re-suspended in fresh PBS buffer. For the *in situ* experiments, after the last centrifugation step NPs@HS were re-dispersed in 100% IgGdHS (total protein concentration 55 mg ml$^{-1}$).

**Physico-chemical characterization of PS and SiO$_2$ NPs.** Bare NPs, Tf adsorbed NPs and HS biomolecular corona NPs were characterized by DCS (see Supplementary Fig. 1, and Supplementary Tables 1–2), DLS (Supplementary Table 2) and NTA (see Supplementary Table 1). DCS measurements were performed using a CPS Disc Centrifuge DC24000. For both PS and SiO$_2$ NPs, a rotational speed of 20,000 r.p.m. and a sucrose density gradient (2–8% for PS and 8–24% for SiO$_2$) in PBS (pH 7.4) were used. The particles were measured between 0.001 and 1 µm, each measurement being calibrated with a standard suitable for the NP density range (PVC standard of nominal size 483 nm in the case of SiO$_2$, and PS standard of nominal size 522 nm for PS NPs) (Analytic Ltd.). The Zetasizer Nano ZS (Malvern Instrument Ltd.) was employed to study the size distribution of NPs. Briefly, 10 µl of the NPs stock was diluted to 1 ml with PBS buffer and measured. A 50 mW laser with a wavelength of 632.8 nm was used as light source and the measurements were recorded at a detection angle of 173° (backscatter). An average of three measurements was recorded at 25 °C for each sample. The concentration of NPs was measured by nanotracking analysis using NanoSight LM10 (Malvern Instrument Ltd.). Briefly 5 µl of the NPs stock was diluted to 1 ml with PBS buffer and measured. The samples were measured for 60 s with manual shutter and gain adjustments. Three measurements of each sample were performed for all NPs. The mean size and standard deviation (s.d.) values obtained by the NTA software correspond to the arithmetic values calculated with the values from all the particles analysed by the software.

**Protein corona analysis.** The protein corona was analysed by SDS–PAGE and mass spectrometry (MS), and protein quantification by BCA determination (see Supplementary Fig. 12 and Supplementary Table 3). The number of Tf molecules adsorbed on PS NPs has been theoretically estimated considering the protein as a rigid sphere of diameter $d = 7.5$ nm and cross-sectional area $\sigma = \pi r^2 = 44.18$ nm$^2$. The maximum number of Tf molecules per NP, corresponding to a close-packed arrangement of the protein on the NP surface[49], is of 2,844 for the 200 nm and 711 for the 100 nm PS NPs, respectively. The concentration of bound Tf and HS corona proteins in the PS and SiO$_2$ NPs surface was calculated with a conventional Thermo Scientific Pierce micro-BCA protein assay, performed according to manufacturer's instructions. The presence of proteins on the NP surface was verified by SDS–PAGE electrophoresis. The NP protein complexes (PS@Tf, PS@Tf/HSA, PS@HS, and SiO$_2$@HS) were analysed after denaturalization by boiling for 5 min the complexes in loading buffer (62.5 mM Tris-HCL pH 6.8, 2% (w/v) SDS, 10% glycerol, 0.01% (w/v) bromophenol blue and 40 mM DTT).

For the MS analysis, the proteins of the corona complexes were first separated of the protein–NP complexes, by 10% SDS–PAGE gel. After running the electrophoresis under constant voltage of 140 V for 10 min, the gel was stained with Commassie blue and the proteins bands taken from each lane before trypsin digestion and mass spectrometry. The gel section containing the proteins was removed using a sterile scalpel and transferred to a clean 0.5 ml sample tube that had been pre-rinsed with acetonitrile. The gel sections were trypsin digested in gel. The samples were re-suspended in 0.1% w/w formic acid before analysis by electrospray liquid chromatography (LC). A HPLC-coupled to a Thermofisher Q-Exactive was used to analyse the samples (LC–MS/MS). Spectra were analysed by label-free quantification using MaxQuant 1.4.1.2 software. A semi-quantitative assessment of the proteins amount was performed by the method of spectral counting (SpC), which represents the total number of the MS/MS spectra for all peptides attributed to a matched protein. The SpC of each protein identity was normalized to the protein mass and expressed as the relative protein quantity by applying the following equation 1:

$$\mathrm{NpSpC}_k = \left( \frac{(\mathrm{SpC}/M_w)k}{\sum_{i=1}^{n}(\mathrm{SpC}/M_w)i} \right) \times 100 \qquad (1)$$

Where $\mathrm{NpSpC}_k$ is the percentage normalized spectral count for protein $k$, SpC is the spectral count identified, and Mw is the molecular weight in kDa for protein $k$.

**QDs synthesis and functionalization.** The synthesis of mercaptopropionic acid protected CdTe quantum dots was performed following the procedure reported by Penades *et al.*[50] A solution of 0.2 mmol of CdCl$_2$ in 40 ml of deoxygenated water was mixed with 0.34 mmol of MPA. The pH was adjusted to 7 with NaOH 1 N and the solution was bubbled with Ar for 30 min. After that, 0.5 ml of freshly prepared NaHTe (40 mM) was added to the three-necked flask and the temperature of the mixture was raised at 140 °C. After 90 min or 20 h stirring at 140 °C, the desired

emission wavelength was obtained (green emitting quantum dots, 530 nm and orange emitting quantum dots, 630 nm), and the reaction was stopped by cooling down to room temperature. The sodium hydrogen tellurite (NaHTe) precursor was freshly synthesized. 0.196 mmol of Te powder was mixed with 0.513 mmol of NaBH$_4$ in 5 ml of deoxygenated water under Ar atmosphere. The reaction was heated to 85 °C under a high flow of Ar and magnetic stirring for 45 min. The QDs were purified by precipitation with acetone. The QDs were separated by centrifugation and dialysed 48 h against PBS buffer. The QDs were characterized using UV–visible absorption spectroscopy and steady-state fluorescence spectroscopy. UV–visible absorption spectra of QD dispersed in PBS were recorded using a Varian Cary 6,000 UV–visible spectrophotometer in a 1 cm path quartz cuvette. Absorbance of 1 ml of different dilutions of QD solution were recorded until optical densities at the excitation wavelength (375 nm) below 0.1 (normally 0.01–0.07) were obtained, to avoid self-absorption effects in the photoluminescence spectra. Photoluminescence spectra of the same samples were recorded between 450 and 700 nm by a Horiba Jobin Yvon Fluorolog-3 spectrofluorimeter at 375 nm excitation wavelength.

The photoluminescence quantum yield measurement procedure where the QY is calculated following the equation 2:

$$QY\, sample = QY\, standard \times \frac{A\, sample}{A\, standard} \times \frac{F\, standard}{F\, sample} \times \frac{n\, sample^2}{n\, standard^2} \quad (2)$$

where $A$ is the integrated area under the emission spectra and $F$ the fraction of the exciting light absorbed at the excitation wavelength, n corresponds with the refractive index of the solvent of the QD and standard solution. As a standard, Rhodamine 6G was used, which has a known quantum yield of 95% in ethanol[51,52].

The QDs were functionalized with antibodies: 1 ml of QDs suspension (0.4 mg, 3 nmol) was mixed with 0.4 mg of EDC and 0.8 mg of Sulfo-NHS in PBS buffer (pH 7.4), and the mixture was incubated at 37 °C for 30 min. The activated QDs solution was purified from the unreacted EDC/Sulfo-NHS by passing it through a PD-10 column using 10 mM PBS as exchange buffer. Then 0.25 mg (1.7 nmol) of IgG antibody was added to the NPs and the mixture was stirred at 37 °C for 1 h. The ratio Ab/QD was optimized to get one antibody per QD. Subsequently, the activated carboxylic groups were blocked with 5 mg of 4-Aminophenyl β-D-galactopyranoside, and the mixture was incubated overnight in a final volume of 2 ml. QDs conjugated with antibodies (1.5 μM) were stored at 4 °C. Non-denaturing polyacrylamide gel electrophoresis (NATIVE-PAGE) was used to check the conjugation of the QDs with the antibody. Photoluminescence spectra were recorded to analyse any variation in the photoluminescence properties of the functionalized QDs. DCS was used to characterize the size distribution and stability of QDAb (Supplementary Fig. 24). Immuno dot blot assay was used to determine the QDAb recognition capacity.

**Biorecognition assay protocol.** For the immunolabelling experiments 100 μl of Tf-adsorbed or 80% HS adsorbed NPs (200 and 100 nm PS or 100 nm SiO$_2$ NPs, see Supplementary Figs 11–14 for the concentration) were incubated with different amounts of immuno-QDs in PBS (pH 7.4) or in IgGdHS, at 37 °C, for 1 h under constant agitation. For the *in situ* experiments, after the incubation with the immunolabels, samples were analysed directly with flow cytometry (see below). In all the other experiments, after the incubation the samples were washed by centrifuging at 20,000 rcf and re-suspending in fresh PBS twice, to remove the unbound immunoprobes. The interaction between the immuno-QDs and the nanoparticles coated with the proteins was studied by differential centrifugal sedimentation (Supplementary Fig. 25), steady-state fluorescence spectroscopy (Supplementary Fig. 26) and flow cytometry.

**Differential centrifugal sedimentation analysis.** The differential centrifugal sedimentation measurements were performed on a CPS Disc Centrifuge DC24000 (CPS Instruments, Inc.). 100 μl of sample were injected in a 2–8% PBS based sucrose gradient for PS NPs, and 8–24% PBS based sucrose gradient for SiO$_2$ NPs. Density values of 1.052 g ml$^{-1}$ and of 1.385 g ml$^{-1}$ were used for the PS and SiO$_2$ NPs, respectively. The rotational speed of the disk was set to 20,000 r.p.m.

**Steady-state fluorescence spectroscopy.** Fluorescence spectroscopy measurements were performed with a Horiba Jobin Yvon Fluorolog-3 fluorimeter using a 45 μl quartz Ultra-Micro cuvette of 3 mm path length (Hellma Analytics). For each sample, two emission spectra were recorded at two different excitation wavelengths, $\lambda$ex = 375 nm and $\lambda$ex = 488 nm, respectively. The first was chosen in the optimal absorption range of the QDs, the second is the wavelength of excitation of the flow cytometer laser and was used to assess the suitability of the QDs for the flow cytometry analysis.

**Flow cytometry.** The side scattering and fluorescence intensity of 50 μl of each nanoparticle–protein complex sample was measured with a BD AccuriTM C6 flow cytometer, at a constant flow rate of 20 μl min$^{-1}$, using a linearly polarized argon-ion laser emitting at 488 nm as excitation source. The beam is elliptically focused to

a cross-sectional area of $10 \times 75\,\mu m^2$ and at the specific flow rate utilized the sample core diameter is of 15 μm. Therefore, the effective beam volume is 1.76 pl. A threshold value of 1,500 in side and forward scattering was applied in all the measurements. Background measurements in PBS buffer were performed before each measurement series. Results are reported as mean of the total fluorescence intensity of nanoparticle-protein complexes labelled with QDAb ± s.d. of several independent replicates ($n = 3$). All the immunolabelling experiment results were analysed with Origin software and GraphPad software. The curves were fitted using the one-site specific binding curve with Hill slope equation.

$$y = \frac{B_{MAX} \cdot x^h}{K_D^h + x^h} \quad (3)$$

NP concentration study by Flow cytometry. Supplementary Figs 4–7 show the variation of the flow cytometry side scattering signal in function of the concentration of NPs for 200 and 100 nm PS@Tf, and 100 and 50 nm SiO$_2$@HS, respectively. By gradually increasing the concentration of NPs analysed, a shift towards higher side scatter values is detected. This can be explained considering that the more NPs are contemporarily present in the detection volume (Fig. 1a), the more the 'average' scattering properties of the suspension change compared with the ones of PBS, until at a certain concentration, which depends on the NP size and properties, the signal of the NPs overcomes the PBS and the background noise. Based on the results in Supplementary Figs 4–7, a final concentration of 0.5 mg ml$^{-1}$ of 200 nm PS NPs and 1 mg ml$^{-1}$ of 100 nm PS NPs and SiO$_2$ NPs were used for all experiments.

**Calibration of the flow cytometry output.** To get semi-quantitative information from the flow cytometry analysis, the flow cytometry signal was calibrated using fluorescence spectroscopy. First, a set of samples of known QD concentration was measured in the spectrofluorimeter to get a calibration curve for the QD signal in fluorescence spectroscopy (Supplementary Fig. 27). Then, PS@Tf NPs titrated with mTfQD$_{630}$ were washed to remove the unbound immunolabels and measured in parallel with both flow cytometry (Fig. 2 in the main text) and fluorescence spectroscopy (Supplementary Fig. 23). The photoluminescence peak of the immunolabelled samples was compared with the QD calibration curve, under the assumption that the total fluorescence of the sample with a certain number of QDs is equivalent to the amount of fluorescence intensity of a solution of known QDs concentration. This allowed us to estimate the number of immuno-QDs on the NP surface. The number of immunolabels per NP for each sample can then be correlated with the corresponding flow cytometry signal (Supplementary Fig. 10). This calibration enables the estimation of the number of exposed epitopes for all the samples titrated with the same QD batch. An analogue calibration procedure was performed for the green emitting QDs (QD$_{530}$).

**Data availability.** Data supporting the findings of this study are available within the article (and its Supplementary Information files) and from the corresponding author upon reasonable request.

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

## Acknowledgements

This work was supported by Science Foundation Ireland (SFI) Principal Investigator Award (agreement no. 12/IA/1422). E.P. and M.C.L.G. acknowledge Science Foundation Ireland (SFI). L.M.H. acknowledges the EU Marie Curie PATHCHOOSER project (PITN-GA-2013-608373). The authors also acknowledge the collaborative EU NANOSOLUTIONS project (NMP-L6-2012-309329). The Conway Institute Flow Cytometry and Proteomics at University College Dublin are also acknowledged. Dr Alfonso Blanco and Prof Matthias Wilm are thanked for assistance with flow cytometry and proteomics experiments.

## Author contributions

M.C.L.G. and E.P. performed all the physico-chemical characterization of nanoparticles and epitope mapping experiments, analysed and interpreted the data. L.M.H. contributed with the synthesis of nanocrystals and supported physico-chemical nanoparticle characterization experiments. E.P. and K.A.D. conceived and designed the experiments. All authors discussed and contributed equally to writing the paper.

## Additional information

**Competing financial interests:** The authors declare no competing financial interests.

**Publisher's note**: 

