## [Peer Review File · Nature Communications]

Reviewers' comments:

Reviewer #1 (Remarks to the Author):

Review of "In situ characterization of nanoparticle biomolecular interactions in complex biological media by flow cytometry" by Lo Giudice et al.

The authors present results in which they study available protein epitopes in protein coronas around polystyrene and SiO₂ nanoparticles. To do so, they use QD-antibodies that can bind to specific epitopes of the proteins of interest and thus are able to measure binding using flow cytometry.

Using this approach, they can quantify the number of epitopes available for the proteins in the corona. They do this for single protein systems, that have been purified (i.e., NPs are spun down and washed to remove excess proteins). They then demonstrate that they can perform the same analysis in a multiplexed format (Tf and HSA at the same time, with red and orange QDAbs). Next, they demonstrate that the approach can be used in situ (i.e., be performed in serum with no centrifugation and separation from the serum). They find loss of affinity for epitope for the in situ measurements, as is expected.

Overall this is an exciting new approach to quantifying the properties of protein coronas, which impact numerous biological applications of nanoparticles. Flow cytometry has not been used to quantify properties of protein coronas around nanoparticles before. This could lead to a greatly enhanced understanding of protein corona properties because it maps available epitopes, not just the presence of a protein, in a corona, and it is the availability of these epitopes that influence how the environment interacts with the nanoparticle-protein corona complex. The authors have performed careful experiments and suitable controls, and also back up their results with independent measurements (bulk fluorescence, gels). Therefore, this manuscript is of interest to the nanotechnology and biotechnology communities, and also to the readership of Nature Communications. There are only some minor issues to be addressed for publication.

- 1) The authors report a number of available epitopes in their measurements. However, how does this number compare to the number of proteins on the NP in the corona? In other words, what percentage of the proteins have epitopes available, and how much is buried?
- 2.) The details in the schematic in Figure 2a are difficult to discern-it would help if the authors could redraw it so that what proteins/antibodies are present are clearer.
- 3.) To overcome the lower size limit of light scattering in flow cytometers, the authors use the "swarm regime" where they measure multiple nanoparticles at a time instead of single particles. They quantitatively estimate that 150 particles are illuminated under these conditions. What is the standard deviation in this estimate?
- 4.) This is more of a stylistic suggestion, but the novelty of using flow cytometry to obtain quantitative information about the corona could be highlighted more. What is the size limitation on the nanoparticles here? One of the potential strengths of flow cytometry over other corona tools such as DSC, FFFF, etc. is that it can probe single particles. While the particles used here are too small for this, larger ones could. Also, with advancements in flow cytometry technologies, how much is this size limitation expected to decrease in the near future? This would help articulate the innovative aspect of this work and the uniqueness of flow cytometry as a tool for studying protein coronas.

Reviewer #2 (Remarks to the Author):

In this work, Dawson et al. report on the use of microfluidic flow cytometry for exploring nanoparticle-biomolecular interactions. They propose that this methodology allows for the detection of molecular motifs presented for biological recognition on the nanoparticle surface directly in biological milieu. Overall, this is a relatively complete and clearly written paper. However, the proposed approach (immunolabeling with quantum dots) is not conceptually new, and the obtained results are not impressive. The choice of Nat. Commun. therefore as a place to publish comes as a bit of surprise. A more specialized journal appears better suitable for this work. Other issues:

1) For the nanoparticle corona complexes preparation, the authors used 32 nM Tf and HSA for incubating with PS nanoparticles. Why use such a low protein concentration? What happens upon increasing the concentration to micromolar range or even higher?

2) The methodology applies to relatively large colloids, as shown in the study (100 / 200 nm), which is actually beyond the size range of most commonly used nanoparticles (less than 100 nm per definition). This raises questions about the versatility of the method to study nano-bio interactions.

3) It has remained unclear to the reviewer how the authors fit the data in Fig. 1 c and d.

4) Scheme 1: "PTMs" unclear (PMTs)?

Reviewer #3 (Remarks to the Author):

Over the past decade, the impact of the immersive biofluid on the biological interactions of natural and engineered nanoparticles has become widely appreciated. Once a nanoparticle enters a biofluid, soluble biomolecules - in particular proteins - adsorb to the particle surface, forming a 'corona' that defines - more or less - the interface between it and the surrounding biological structures (for e.g. cells). The biomolecular corona is therefore a critical determinant of the subsequent biological interactions of nanoparticles and their ultimate fate within a biological system.

Dawson and colleagues have pioneered the study of biomolecule-nanoparticle interactions within complex biological fluids since their seminal studies in the mid-2000s. In this submission, they describe the development and validation of a methodology that addresses a long-standing question in the field: which cell receptor-binding motifs within an adsorbed (or functionalized) protein layer are accessible to interact with their cognate receptors? Many proteins with a propensity to interact with nanoparticles - including vitronectin, fibronectin, immunoglobulins, and transferrin - contain well-characterized motifs that are recognized by specific cell-surface receptors. However, whether these motifs are freely accessible for interaction or 'buried' within the adsorbed biomolecule layer will determine which cell types a nanoparticle will interact with and the consequence of those interactions.

In an elegant application of a widely-available piece of laboratory equipment, Dawson and his colleagues used flow cytometry to quantify the number of transferrin (Tf) and IgG receptor-binding motifs that are accessible to recognition probes following adsorption to polystyrene or silica beads. The probes took the form of quantum dots functionalized with antibodies that specifically recognized cell-binding motifs in Tf and IgG.

Previous strategies to characterize the accessibility of molecular motifs within the adsorbed protein layer have first purified the nanoparticle target from excess unbound biomolecules. This strategy is

problematic as it can result in the loss or disruption of the adsorbed protein layer - particularly weakly adsorbed biomolecules - leading to conclusions that are not physiologically relevant. The methodology of Giudice et al. is the first that offers the potential to probe the accessibility of motifs *in situ* - without requiring this purification step. Indeed, one of the most interesting observations from this study is that once a nanoparticle plus its bound protein layer is introduced into a physiologically-relevant biofluid (in this case blood plasma) the accessibility of the Tf and IgG cell-binding motifs is significantly inhibited. The authors suggest that this is due to additional adsorbed biomolecules from plasma 'covering' the motifs.

The evaluation of protein-nanoparticle interactions within complex biofluids is a necessary step in order to develop nanoparticle formulations for medical applications that perform as intended with minimal off-target or unexpected biological interactions. It is also a critical aspect to toxicological screening and understanding the interactions of natural nanoparticles within biological systems. While the 'in situ' characterization of nanoparticles within physiological biofluids - as accomplished by Dawson and colleagues in this report - is technically challenging, it is absolutely vital if we are to draw meaningful conclusions and develop effective formulations. The strategy described in this paper is an important step in this direction and adds a much-needed tool to the repertoire of nanobioscientists. Moreover, since it does not use specialized equipment, it can be widely-applied and extended by many researchers across laboratories. This report is therefore of significant interest and will be widely read by the diverse communities interested in nanoparticle interactions with biological systems. The concept is thus suitable for publication within a premier journal with a diverse readership such as *Nature Communications*.

However, in its current form, the manuscript is not yet suitable for publication. Several key control experiments and a broader discussion of the potential and limitations of the approach are missing. These elements are important to ensure the fidelity of the results and conclusions presented herein and so that this report may serve as a guide to others. These issues and suggested improvements are discussed in more detail below. Once they are fully addressed, the manuscript should be re-considered for publication.

Specific Issues/Recommendations

1) A key observation is the reduction in QD probe binding to target particles in the presence of plasma. However, could this be a result of biomolecules in plasma interfering with recognition of target epitopes by antibody-grafted QDs? In other words, might biomolecule interactions with the Ab-grafted QDs themselves prevent interaction with the target epitope? This issue should be experimentally addressed. One strategy may be to perform an ELISA whereby antibodies against QD-bound Abs are used to quantify the availability of QD-bound Abs in the presence of excess plasma. Alternatively, one could expose the quantum dots to plasma, purify them, and then measure their association with target particles.

2) In Supplementary Figure 6, QDs modified with antibodies appear not to be moving in the gel, which may be a result of aggregation. Differences in the aggregation state of the QDs in plasma/serum could impact the fluorescent readout and recognition of the target epitope. The aggregation state of the QD probes in the various biological media should be characterized (by dynamic light scatter, size exclusion chromatography, or equivalent) and reported. The contribution of potential differences in aggregation state in terms of binding avidity and fluorescence readout in flow cytometry should be discussed.

3) Non-specific binding of QD probes to exposed target particle surface is a potential source of the observed fluorescence signal. Binding sites may be 'blocked' in the presence of excess plasma, which could lead to the observed decrease in probe binding. Supp Fig 13 describes non-specific binding experiments using a BSA-modified QD. However, BSA and antibodies (the molecule used for biorecognition studies) are not similar proteins in size or chemical properties. This study should be repeated to include a non-specific antibody as control to ensure that binding is the result of specific association of the QDs with target proteins.

4) Even plasma depleted in IgG or Tf may have a significant concentration of remaining IgG or Tf relative to the QD probe concentration. These residual biomolecules represent an off-target population that would block QD probe binding sites. Thus, the reduction in bound population of QDs in Fig 4 may equally be a result of the presence of an off-target competing population of free IgG/Tf residual in the depleted serum. Control experiments to characterize the residual IgG/Tf content in plasma are required. This was partially verified in Supplementary Fig 5b. However, it appears that 'Tf depleted human plasma' was not characterized for the content of residual Tf by Western blot in Supplementary Figure 5b).

5) A wider discussion of the applicability and limitations of the described approach is needed. Some important questions to address include: a) How does the applicability of flow cytometry for this application depend on target nanoparticle properties? In particular size, shape, charge, and composition. Flow cytometry is intended for the analysis of cells and not nano-sized materials. b) Can this approach be used with fluorescently-labeled antibodies to avoid the need for QDs? c) What is the lower limit of detection of available epitopes on the target nanoparticle surface? d) What effects does shear stress have on the protein corona under flow? e) How do you differentiate aggregated from dispersed particles (either target particle or QD probe) using this approach?

Reviewer #1 (Remarks to the Author):

1) The authors report a number of available epitopes in their measurements. However, how does this number compare to the number of proteins on the NP in the corona? In other words, what percentage of the proteins have epitopes available, and how much is buried?

For a broader range of systems, our developing experience suggests that the number of available epitopes for hard corona layers or other grafted biomolecules is typically between 10 % - 30 %, more usually at the much lower end of this range. Here, for this single protein (Transferrin, Tf) one can carry out all of the analyses very precisely, and monolayer composition on the nanoparticle surface was calculated theoretically and determined experimentally by BCA analysis. These data are reported in the Supplementary Figure 3. The percentage of exposed Tf epitopes, semi-quantitatively determined by fluorescence spectroscopy and flow cytometry, corresponds to the 24 % of the total amount of protein adsorbed (reported by BCA) for 200 nm PS NP and 30 % for 100 nm PS NP. When the analysis of the exposed epitopes was performed *in situ* by flow cytometry (Fig. 3), only about 5 % of the total amount of epitopes are detected after the particles are dispersed in complex media. Therefore around 80 % of the epitopes labelled in the model system in PBS are buried *in situ*. We have added the percentage of exposed epitopes to the new version of the article (lines 126-128, 174 and line 203 and 207).

For more complex (e.g. serum) systems, the percentage of available epitopes is more challenging to calculate, due to the lack of experimental techniques able to provide quantitative information on the amount of each protein of the biomolecular corona. Nevertheless, combining mass spectrometry analysis and micro-BCA assay (as described in Supplementary Figure 3) we can approximately estimate the total amount of a given protein per NP. For IgG total number of proteins is typically between 15 % - 20 % for polystyrene NP and silica NP, as a proportion of the total proteins of the biomolecular corona. Therefore, from Figure 4 the amount of exposed IgG epitopes corresponds to about 50 % for polystyrene NP and 85 % for silica NPs of the total IgG (between 5 % - 10 % of epitopes detected *in situ*). Variations of signals from bound QDAb derived from the sample preparation

(nanoparticle, nanoparticle-protein complexes, mixing, etc.) are typically small, the accuracy of the estimated numbers depends on errors accumulated in the calibration process.

2) *The details in the schematic in Figure 2a are difficult to discern-it would help if the authors could redraw it so that what proteins/antibodies are present are clearer.*

We have redrawn and enlarged the Figure 2a.

3) *To overcome the lower size limit of light scattering in flow cytometers, the authors use the "swarm regime" where they measure multiple nanoparticles at a time instead of single particles. They quantitatively estimate that 150 particles are illuminated under these conditions. What is the standard deviation in this estimate?*

This is an interesting question that depends on the particular specifications of the device used. In general the fluctuations within a volume (as a proportion of the mean) increase in accordance with use expectations from the central limit theorem. However, under this kind of sheath-like microfluidic flow there are other considerations, and the fluctuations are not completely understood. It will be useful in future to understand this more, those fluctuations could contain also interesting information, such as variability of epitope exposure among the NP population. The actual estimation of the average swarm size has been carried out following the procedure described in the text (materials and methods and scheme 1). More specifically, considering the specifications for the optics (laser profile 10 x 75 μm) and fluidics (flow cell 200 μm) for the instrument BD Accuri™ C6 Flow Cytometer, the conditions to perform the experiments (core size of 15 μm) and following a method described by Van der Pol. E. et al. we calculate the volume illuminated by the laser during the acquisition time. The concentration of nanoparticles is determined experimentally by nanotracking analysis (NTA analysis), therefore and for effective volumes of that size, we find an average of 150 NP with experimental deviations between 80 and 220 NPs. Furthermore, taking the coefficient of variation (CV) of the flow cytometry event distribution as statistical assessment of the peak spread, we find fluctuations in numbers between 100 and 225 NPs. As the field of swarm and particle-by-particle measurements advances all of those questions will need to be considered more carefully.

4) *This is more of a stylistic suggestion, but the novelty of using flow cytometry to obtain quantitative information about the corona could be highlighted more. What is the size limitation on the nanoparticles here? One of the potential strengths of flow cytometry over other corona tools such as DSC, FFFF, etc. is that it can probe single particles. While the particles used here are too small for this, larger ones could. Also, with advancements in flow cytometry technologies, how much is this size limitation expected to decrease in the near future? This would help articulate the innovative aspect of this work and the uniqueness of flow cytometry as a tool for studying protein coronas.*

Here, in order to deal with all types of nanoparticles (fluorescent and non-fluorescent), multiple nanoparticles are simultaneously illuminated by the laser and captured within the detection volume (termed the "swarm regime"), and the outcome counted as one single event. By using routine instruments as the one employed in this work, which presents limit of detection of 500 nm, we were able to characterize swarms of NPs in the size range of 50 nm. However, we are currently investigating and developing microfluidics approaches that allow even for particle-by-particle evaluation. Commercial instruments are also being developed with more advanced lasers, robust photodetectors, and indeed combining with imaging or other techniques such as mass spectrometry (mass cytometry). These novel systems allow to decrease the size limitation and they are capable of detecting fluorescence from very small groups of (in some cases even individual) particles (decorated with reporter binder such as immuno-QDs), coupled with simultaneous measurements of low angle (forward) and high angle (side) light scattering. It is also worth noting (as above) that the price for more individual particle treatments is that the numbers in the population that can be measured decreases quite significantly, and then the value of that approach has to be weighed against the nature of the population, and the numbers required for an analysis. Almost certainly the answer lies in being able to tune between larger swarms into the regime where one can analyze the fluctuations

(see above) in a fundamental manner and thereby obtain maximum information on the nature of the ensemble. We implemented the discussion of the manuscript taking into consideration the suggestions highlighted by the reviewer.

Reviewer #2 (Remarks to the Author):

1) For the nanoparticle corona complexes preparation, the authors used 32 nM Tf and HSA for incubating with PS nanoparticles. Why use such a low protein concentration? What happens upon increasing the concentration to micromolar range or even higher?

This is now explained more clearly in the text. The protein concentration used (64 nM , $5 \text{ mg} \cdot \text{mL}^{-1}$) for Transferrin and a ratio one to one molar for the two-protein model (32 nM of both proteins) to form the NP protein complexes is above the saturating concentration of protein for this size of NP. This has been previously measured and reported (Pitek A. *et al. Plos one*, 2012 and Kelly P.M. *et al. Nat. Nanotech.* 2015) and indeed we find no differences for higher concentrations. See details from one of our previous publications (Figure S3 from Kelly P.M. *et al.*) copied below.

Comparison of three techniques used to investigate the amount of protein attached to the polystyrene nanoparticles. The results indicate that the polystyrene surface is completely covered by proteins as verified by three independent techniques (Differential centrifugal sedimentation DCS, protein concentration by BCA assay and SDS-PAGE). (a) DCS shows a shift in apparent diameter (black), mass of protein attached per mg of PS nanoparticles (red), determined using a micro bicinchoninic acid assay (μ BCA) protein assay. Densitometry of the Transferrin protein bands observed in an SDS-PAGE gel was carried out using image J; (b) SDS-PAGE gel showing the proteins attached to PS nanoparticles as a function of increasing Transferrin concentration. The same concentration of nanoparticles for each condition was loaded. Bands were visualized using a Coomassie blue stain.

2) The methodology applies to relatively large colloids, as shown in the study (100 / 200 nm), which is actually beyond the size range of most commonly used nanoparticles (less than 100 nm per definition). This raises questions about the versatility of the method to study nano-bio interactions.

We chose those sizes 200 nm and 100 nm nanoparticles as a model system to set up the platform and to compare with previous studies (Kelly *et al. Nature Nanotech.* 2015). Moreover, as outlined in the supporting information,

in the swarm regime, by varying the concentration of nanoparticle utilized it is possible to reach a threshold above which the signal of the nanoparticles overcomes the instrumental noise (see Supplementary Figures 11, 12, and 13). Working in the swarm regime, size is not a substantial constraint and depends more on the system details (concentration, scattering and fluorescence properties of the nanoparticle of interest). For interest, we have now included in Supplementary Figure 14 an example of the applicability of the system to detect nanoparticles in the 50 nm size range.

NP concentration analysis on flow cytometry for fluorescence 50 nm SiO₂ NPs (FTIC). Overlap of the side scattering distribution of PBS (background noise) and SiO₂ NPs samples at different concentrations. Histograms overlap of the side scattering distribution of increasing concentration of SiO₂ NPs (left), and of the fluorescence signal in the green channel (530/30 nm) (right)

3) It has remained unclear to the reviewer how the authors fit the data in Fig. 1 c and d.

It is indeed worth remarking on that. The immuno-QDs binding curves were fitted using the one-site specific binding curve with Hill slope equation

$$y = \frac{B_{MAX} \cdot x^h}{K_D^h + x^h}$$

where B_{MAX} is the maximum specific binding in the same units as y , K_D is the immunoprobe concentration needed to achieve a half-maximum binding at equilibrium and h is the Hill slope (related with the cooperativity of the binding). The fitting parameter B_{MAX} is used to estimate the maximum value of exposed epitopes in the examples presented. The fitting used in the analysis has been clarified in the manuscript.

4) Scheme 1: "PTMs" unclear (PMTs)?

The typo has been corrected.

Reviewer #3 (Remarks to the Author):

1) A key observation is the reduction in QD probe binding to target particles in the presence of plasma. However, could this be a result of biomolecules in plasma interfering with recognition of target epitopes by antibody-grafted QDs? In other words, might biomolecule interactions with the Ab-grafted QDs themselves prevent interaction with the target epitope? This issue should be experimentally addressed. One strategy may be to perform an ELISA whereby antibodies against QD-bound Abs are used to quantify the availability of QD-bound Abs in the presence of excess plasma. Alternatively, one could expose the quantum dots to plasma, purify them, and then measure their association with target particles.

This is an interesting question that we did consider in several different ways. Specific interactions between probe (QDAb) and target protein are of course eliminated in the experiments by depletion of the target from the media (IgG in the example presented). To eliminate unspecific interactions of the carboxylic groups present on the QDs surface with the biomolecules of the media we modify the surface of our QDs with a suitable blocking agent (4-Aminophenyl β -D-galactopyranoside) after the antibody immobilization (reported in the methods section). Those steps are important in developing the system. The recognition capabilities of the immuno-QDs with the target molecules present in complex biological media and in the protein corona on the NP have been confirmed by dot blot analysis (Supplementary Figure 4 has been modified in order to include more results). More specifically, we have checked the dot blot for the interaction of the immuno-QDs with the target, in PBS, 50 % of human serum and 50 % of IgG depleted human serum. In the case of the IgG depleted serum, the biomolecules in solution do not interfere with recognition by the immuno-QDs (Figure c) and immuno-QDs interaction decreases when the percentage of the IgG depleted serum is increased (Figure d).

Immuno dot blot assay. a) Immuno dot blot for monoclonal antibody anti-HSA (1) and monoclonal anti-Tf (2). Tf and HSA were spotted on each of the PVDF membranes at two different concentrations, 1 mg·mL⁻¹ (top) and 0.2 mg·mL⁻¹ (bottom). b) Immuno dot blots of monoclonal antibody against IgG (Fc) (1) and against ApoB-100 (2).

The membranes were spotted with HSA as a negative control (45 mg·mL⁻¹, A), HS (77 mg·mL⁻¹, B), delipidised serum (58 mg·mL⁻¹, C) and IgG depleted HS (55 mg·mL⁻¹, D). c) Immuno dot blot analysis of mIgGQD₆₃₀. Human serum have been spotted on the PVC membrane. The membranes are incubated with a solution of mIgGQD₆₃₀ in PBS (1), 50 % human serum (2) and 50 % IgG depleted human serum (3) d) Immuno dot blot analysis of IgG (Fc) recognition by immuno-QDs (mIgGQD₆₃₀) incubated in 10 % or 50 % IgG depleted HS. mIgGQD₆₃₀ have been spotted on the PVC membrane. Afterwards the membranes were incubated with the 100 nm fluorescent SiO₂@HS NPs in PBS, 10 % and 50 % IgG depleted HS. The IgG epitope recognition is decreasing proportionally to the amount of proteins in the media.

In addition we investigated the exposure of IgG epitopes on PS NPs in PBS and *in situ* (in IgG depleted serum) using a different immunoprobe, consisting of Rhodamine conjugated IgG antibody (see Figure below, now included in supplementary figure 24). It is shown that it leads to the same broad outcome as suggested by the QDAb, i.e. a strong reduction of the interactions in complex biological media occurs. This suggests us that the reduction of the labeling capabilities of the immuno-QDs in complex media are not due to the interfering interactions between QDs and other proteins from the media.

2) In supplementary Figure 6, QDs modified with antibodies appear not to be moving in the gel, which may be a result of aggregation. Differences in the aggregation state of the QDs in plasma/serum could impact the fluorescent readout and recognition of the target epitope. The aggregation state of the QD probes in the various biological media should be characterized (by dynamic light scatter, size exclusion chromatography, or equivalent) and reported. The contribution of potential differences in aggregation state in terms of binding avidity and fluorescence readout in flow cytometry should be discussed.

We considered this interesting issue in developing the system, and it is worth mentioning some of that background. It is of course important to characterize and understand the aggregation state of the immuno-QD probes. In fact the construction of these probes is not a simple issue, and (see above) considerable effort was dedicated to reach the current design. Indeed, we learned quite a bit by looking at different variants illustrating how aggregation and non-specific effects show up for those different designs in outcomes ranging from scattering in flow cytometry and fluorescence. The QDAb can be characterized by a variety of techniques, including DCS, fluorescence spectroscopy and flow cytometry. In Supplementary Figure 6e we show the photoluminescence spectra of the immunoprobes QDAb in human serum over time. No substantial variation in the intensity of the photoluminescence emission peak was observed in any case. Regarding the electrophoretic mobility shown in Supplementary Figure 6d, the fact that the QDAb do not move further in the gel is a consequence of time chosen

for the experiment, short enough to be able to differentiate between free QD, free Ab, and conjugates. Increasing the running time or modifying the pore size of the gel also allows all three to move, including the QDAb. To address that point new picture of an agarose gel (larger pore size) is included in Supplementary Figure 6.

The covalent immobilization of the antibody onto the QD is characterized by the QD electrophoretic mobility in a 1 % agarose gel (50V, 30 min). QDs control (QDs activated with EDC and Sulfo-NHS and blocked with 4-Aminophenyl β -D-galactopyranoside) (1), QDsAb with different ratios of Ab conjugated per QD have been analyzed (2-4). 0.6 Ab/QD is the ratio used for all the experiment (3) and QDs free (5).

We also report more characterization of the QDs in biological media using differential centrifugal sedimentation. DCS (see graph below) suggests that the QD-Ab are not aggregated in human serum (similar size distribution to QDs-Ab in PBS). This data has been now included in the Supplementary figure 7.

It is also worth mentioning that flow cytometry is a rather convenient tool in that regard, and we observed that side scattering (due to the small size under consideration) quite sensitively detects aggregation. Side scattering and fluorescence distributions for the solution of QDAb dispersed in PBS or biological media (human serum) are shown in Supplementary figure 19.

Immuno-QDs characterization by DCS: Size distribution of mIGQD in PBS and in 50 % of human serum.

3) Non-specific binding of QD probes to exposed target particle surface is a potential source of the observed fluorescence signal. Binding sites may be 'blocked' in the presence of excess plasma, which could lead to the observed decrease in probe binding. Supp Fig 13 describes non-specific binding experiments using a BSA-modified QD. However, BSA and antibodies (the molecule used for biorecognition studies) are not similar proteins in size or chemical properties. This study should be repeated to include a non-specific antibody as control to ensure that binding is the result of specific association of the QDs with target proteins.

BSA is a fairly standard choice of blocking agent in most of the immunoassays, and we believe it functions well here too. We have included in Supplementary Figure 13 a non-specific antibody used as a control to ensure the specificity in the binding by the immuno-QDs with the target molecule. Similar results were obtained with the antibody control and the BSA.

The specificity of the immuno-QD binding has been determined by studying the interaction of the PS@Tf NPs with different concentration of QDs functionalized with a new immunoprobe consists of the same QDs functionalized with an antibody control (monoclonal antibody anti-Apolipoprotein E).

4) Even plasma depleted in IgG or Tf may have a significant concentration of remaining IgG or Tf relative to the QD probe concentration. These residual biomolecules represent an off-target population that would block QD probe binding sites. Thus, the reduction in bound population of QDs in Fig 4 may equally be a result of the presence of an off-target competing population of free IgG/Tf residual in the depleted serum. Control experiments to characterize the residual IgG/Tf content in plasma are required. This was partially verified in supplementary Fig 5b. However, it appears that 'Tf depleted human plasma' was not characterized for the content of residual Tf by Western blot in supplementary Figure 5b).

The Tf depletion is frequently prepared in the laboratory, and residual Tf is always characterized by Western Blot and ELISA, as previously reported by Salvati A. et al. Nat. Nanotechnology (2013) (see Figure S8 from Salvati A. et al.). For this work, Tf depleted plasma and IgG depleted serum were western blotted for residual protein, and the results are now included in Supplementary Figure 5 (copied below).

Preparation and characterization of Tf depleted human plasma and IgG depleted human serum. a) Human plasma depleted of endogenous Tf was prepared by affinity chromatography using HiTrap NHS-Activated HP columns (GE Healthcare) activated with polyclonal antibody anti human Tf, following the protocol described in Salvati et al. The Tf depleted human plasma has been characterized by SDS-PAGE and Western Blot. For Western blot immuno-detection of residual Tf in the depleted plasma, the plasma before and after depletion (together with the proteins recovered in the corresponding eluted fractions as controls) were separated by SDS-PAGE (10 % polyacrylamide gels). b) Commercial IgG depleted human serum was characterized by SDS-PAGE and Western blot.

5) A wider discussion of the applicability and limitations of the described approach is needed. Some important questions to address include: a) How does the applicability of flow cytometry for this application depend on target nanoparticle properties? In particular size, shape, charge, and composition. Flow cytometry is intended for the analysis of cells and not nano-sized materials. b) Can this approach be used with fluorescently-labeled antibodies to avoid the need for QDs? c) What is the lower limit of detection of available epitopes on the target nanoparticle surface? d) What effects does shear stress have on the protein corona under flow? e) How do you differentiate aggregated from dispersed particles (either target particle or QD probe) using this approach?

Those are all substantive and important questions for the future (the question of particle size is dealt with above), and we can give a brief summary of our observations and conclusions to date. Our basic idea is that there are now compelling arguments to suggest that we must understand the molecular presentation on the surface of nanoparticles to be able to make contact with the biological impact. The targeting capabilities of engineered nanoconstructs critically depend on the orientation and exposure of the particular molecular moieties that can be

drastically screened *in situ*, as a consequence of being in biological media. The general approach of microfluidic particle analysis is not affected by the shape or other nanoparticles' properties but most certainly the outcome of epitope availability is. Indeed, even preparative details of particular systems (for instance the grafting synthetic strategy) can affect the outcome. This is not usually taken into account in nanomedicine using synthetic nanoparticles, but there is a strong motivation to be sure that any failures of particular targeting concepts are not solely a consequence of the particle design leading to limited access of targeting moieties.

Here we present flow cytometry as a high-throughput analytical tool which offers the possibility to gain information on the *in situ* determination of the particular molecular moieties available on the NP surface. Simultaneous measurements of low angle (forward) and high angle (side) light scattering allow to discriminate between different nanoparticle subpopulations (aggregated from dispersed nanoparticles), providing simultaneous information on the local state of dispersion, and the fluorescence per particle (examples of aggregated NP samples are provided in Supplementary Figure 16).

The use QDs constructs as reporter binders offers valuable assets (broad absorption spectra and narrow emission bandwidths, high absorption cross section, brightness, and photostability) but in principle dyes could be used also. We have found that largest single advantage of using QD constructs arises when we multiplex and detect larger numbers of epitopes simultaneously. This is a very powerful tool to characterize the surface of nanoparticles in a biological context.

The question of particle shear stress under flow is also of great interest for it could be important in real biological scenarios such as blood circulation. However, with this regime of flow rates and conditions we expect (and find) no evident effects.

We implemented the discussion of the manuscript taking into consideration the suggestions highlighted by the reviewer.

REVIEWERS' COMMENTS:

Reviewer #1 (Remarks to the Author):

The authors have sufficiently addressed all of the comments raised in the original review. Therefore, publication of the manuscript is recommended.

Reviewer #2 (Remarks to the Author):

The authors have addressed the specific issues that I had raised in the previous round of reviews in a satisfactory manner.

Reviewer #3 (Remarks to the Author):

In their rebuttal and revised manuscript, the authors have done a quite thorough job of addressing all of the specific issues that I raised in my original review. I am now happy to give a full and hearty endorsement of this manuscript for publication in Nature Communications. I believe, as I did upon first reading it, that it is an important contribution to the field of nanobiotechnology. In its revised form, I believe that this manuscript will be well-read, illuminating, and highly cited.